# Differentiating Multiple Myeloma and Osteolytic Bone Metastases on Contrast-Enhanced Computed Tomography Scans: The Feasibility of Radiomics Analysis

**DOI:** 10.3390/diagnostics13040755

**Published:** 2023-02-16

**Authors:** Seungeun Lee, So-Yeon Lee, Sanghee Kim, Yeon-Jung Huh, Jooyeon Lee, Ko-Eun Lee, Joon-Yong Jung

**Affiliations:** 1Department of Radiology, Seoul St. Mary’s Hospital, College of Medicine, The Catholic University of Korea, Seoul 06591, Republic of Korea; 2Department of Radiology, Yeouido St. Mary’s Hospital, College of Medicine, The Catholic University of Korea, Seoul 07345, Republic of Korea; 3Department of Biostatistics and Data Science, School of Public Health, The University of Texas Health Science Center at Houston, Houston, TX 77030, USA

**Keywords:** multiple myeloma, neoplasm metastasis, multidetector computed tomography, diagnosis, algorithm, radiomics

## Abstract

Osteolytic lesions can be seen in both multiple myeloma (MM), and osteolytic bone metastasis on computed tomography (CT) scans. We sought to assess the feasibility of a CT-based radiomics model to distinguish MM from metastasis. This study retrospectively included patients with pre-treatment thoracic or abdominal contrast-enhanced CT from institution 1 (training set: 175 patients with 425 lesions) and institution 2 (external test set: 50 patients with 85 lesions). After segmenting osteolytic lesions on CT images, 1218 radiomics features were extracted. A random forest (RF) classifier was used to build the radiomics model with 10-fold cross-validation. Three radiologists distinguished MM from metastasis using a five-point scale, both with and without the assistance of RF model results. Diagnostic performance was evaluated using the area under the curve (AUC). The AUC of the RF model was 0.807 and 0.762 for the training and test set, respectively. The AUC of the RF model and the radiologists (0.653–0.778) was not significantly different for the test set (*p* ≥ 0.179). The AUC of all radiologists was significantly increased (0.833–0.900) when they were assisted by RF model results (*p* < 0.001). In conclusion, the CT-based radiomics model can differentiate MM from osteolytic bone metastasis and improve radiologists’ diagnostic performance.

## 1. Introduction

Metastatic cancer is the most common malignant tumor in bones, followed by multiple myeloma (MM). Many patients with advanced cancer suffer from bone metastasis. Bone metastasis is not uncommon as an early manifestation before a primary tumor is detected. Metastasis can be diagnosed by imaging modalities, including radiography, computed tomography (CT), magnetic resonance imaging (MRI), bone scan, and fluorine-18 fluorodeoxyglucose positron emission tomography/computed tomography (^18^F-FDG PET/CT) [1,2]. In general, bone metastasis is divided into osteolytic, osteoblastic, and mixed types. Osteolytic metastasis is the most common form of metastasis, appearing as purely radiolucent lesions. Osteoblastic metastasis is common in prostate and breast cancers and is seen as hyperdense lesions on CT images and radiographs due to osteoblastic activity. The mixed type includes both osteolytic and osteoblastic features. Bone metastases from the same primary cancer do not always show the same type, and the types can vary among lesions in a single patient. Moreover, chemotherapy or radiation therapy can alter the findings [3].

MM is the second-most common skeletal malignancy after metastasis [4]. MM is a hematologic malignancy and is included in the disease spectrum of monoclonal gammopathy [5]. MM is diagnosed when myeloma biomarkers are elevated, plasmacytoma is present, or there is end-organ damage (i.e., CRAB criteria: hypercalcemia, renal insufficiency, anemia, and bone lesions) [5]. Radiographic imaging plays an important role in the diagnosis of bone lesions. The International Myeloma Working Group (IMWG) proposes low-dose whole-body CT as the primary diagnostic imaging modality for MM [6]. Radiography, MRI, and PET/CT can also help diagnose and evaluate treatment response in MM [5,6]. Common imaging findings in MM include well-defined osteolysis without periosteal reaction, expansile radiolucent lesions, endosteal scalloping, extramedullary masses, diffuse osteopenia, and pathologic fractures [7]. Osteolytic bone metastases have similar imaging findings. While establishing an accurate diagnosis is essential for proper treatment, it is often challenging to differentiate MM and osteolytic bone metastases.

Our previous research, which attempted to differentiate MM and osteolytic metastases on contrast-enhanced CT images, found that intratumoral and intertumoral homogeneity were key image findings for MM [8]. Another study that performed histogram analysis of diffusion-weighted MRI scans revealed that the apparent diffusion coefficient was more narrowly distributed for MM than for metastases [9]. In addition, several studies using CT and MR images confirmed the difference in homogeneity between MM and bone metastases. Mutlu et al. revealed that osteolytic bone metastases showed more high-density areas and more significant heterogeneity than MM [10].

Radiomics analysis allows objective evaluation of tumor heterogeneity and could provide new information for the differentiation of tumors based on imaging [11,12,13]. A recent MRI radiomics study by Xiong et al. reported that a machine-learning model developed from radiomics features of T1- and T2-weighted images successfully differentiated MM and osteolytic bone metastases [14]. An investigation of ^18^F-FDG PET/CT images found that the radiomics model successfully classified MM and bone metastases with significantly improved diagnostic performance compared to both human experts and the conventional PET parameter SUV_max_ [15]. However, a machine learning model using CT radiomic features can be more beneficial for differentiating bone metastases and MM, as many patients undergo thoracic, abdominal, and spinal CT more frequently than spinal MRI because of easy access.

We hypothesized that a machine learning model using the input of radiomic features from CT images could differentiate MM lesions and osteolytic bone metastases. Therefore, the purpose of this study was to evaluate the diagnostic performance of a radiomics model that differentiates MM lesions and osteolytic bone metastases using CT images.

## 2. Materials and Methods

### 2.1. Patients

The institutional review boards of Seoul St. Mary’s Hospital (institution 1) and Yeouido St. Mary’s Hospital (institution 2), Seoul, Republic of Korea, approved this retrospective study and waived the need for informed consent. All patients were enrolled from institutions 1 and 2. Eligible individuals included consecutive adult patients (at least 17 years old) histopathologically diagnosed with either MM or primary cancer. All patients had undergone contrast-enhanced CT imaging of the chest, abdomen, and spine between 2014 and 2020 prior to cancer-related treatment. All patients had at least one osteolytic bone tumor without an osteoblastic component. We excluded patients with concurrent MM and another primary cancer, a history of spine surgery, pathologic fractures, or lesions too small to be evaluated (maximum diameter less than 1 cm).

A radiologist (S.L., with 3 years of experience in musculoskeletal radiology) and a one-year resident in training (Y.-J.H.) reviewed the medical records and CT images of the included patients in unity. They selected up to three lesions per patient by consensus according to the priority of lesion diameter. When a consensus could not be reached, another radiologist (S.-Y.L., with 12 years of experience in musculoskeletal radiology) made the final decision for lesion selection. In the process of selecting target lesions, the reviewers were blinded to the clinical information. Patients from institutions 1 and 2 were assigned to the training set and external test set, respectively (Figure 1).

### 2.2. CT Data Acquisition

Contrast-enhanced CT imaging was performed at the two institutions. Several scanners were used at institution 1: a 128-row multidetector CT scanner (Somatom Definition AS+, Siemens Healthineers, Erlangen, Germany) and 64-channel CT scanners (Somatom Sensation 64, Siemens Healthineers; Discovery CT750 HD, GE Healthcare, Chicago, IL, USA), along with iodine contrast media (Iobrix 300, Taejoon Pharm, Seoul, Republic of Korea). A 256-channel CT scanner (Brilliance iCT, Philips Medical Systems, Eindhoven, The Netherlands) and iodine contrast media (Bonorex 350, Central Medical Service, Seoul, Republic of Korea) were used at institution 2. The CT parameters were as follows: 100–120 kVp tube voltage, 100–200 mAs tube current under automatic modulation, and 3–5 mm section thickness. A filtered-back projection algorithm with B30, B31, or standard kernels was used for reconstruction.

### 2.3. Volume of Interest (VOI) Segmentation and Radiomics Feature Extraction

The entire volume of interest (VOI) was manually segmented based on contrast-enhanced CT images. Segmentation was performed using an open-source program (ITK-SNAP software, version 3.8.0, http://www.itksnap.org, accessed on 1 December 2020) (Figure 2) [16]. To correct the variability of voxel size, segmented VOIs were resampled to isometric voxels of 1 mm × 1 mm × 1 mm [17]. The radiomics features of VOI segmentation were extracted using the PyRadiomics package (https://github.com/Radiomics/pyradiomics/, accessed on 10 June 2021) [18]. VOI segmentation was performed by one trained technologist (K.-E.L., with 2 years of experience in medical imaging segmentation). A radiologist (S.-Y.L.) supervised the segmentation.

### 2.4. Radiomics Feature Reduction and Selection

Before the machine learning model was developed, radiomics features were selected in two steps to avoid dimensionality issues. First, we evaluated the reproducibility of VOI segmentation [19]. One radiologist (S.K., with 1 year of experience in musculoskeletal radiology) randomly selected 30 lesions from the study cohort, independently completed VOI segmentation on the CT images, and extracted the radiomics features in the same manner as described above. The inter-examiner agreement between 2 radiomics features extracted from individual segments by the 2 examiners was evaluated using the intraclass correlation coefficient (ICC) [20]. Radiomics features with ICC > 0.75 were selected for model construction [21]. Second, redundant radiomics features were reduced by using a binomial elastic net. The elastic net removed all features that correlated with each other [22]. The number of features to be selected was determined by tuning optimal α and λ values using 10-fold cross-validation.

### 2.5. Radiomics Model Development

A random forest (RF) classifier algorithm was built using commercially available software (mlr package, R statistical software version 3.5.2; R Foundation for Statistical Computing, Vienna, Austria; available at https://cran.r-project.org/bin/windows/base/old/3.5.2, accessed on 10 June 2021). The generalization capacity of the model was evaluated by 10-fold cross-validation. The feature importance for all cross-validation experiments was calculated based on the mean decrease in node impurities from the developed model as measured by the Gini index. The RF models were developed in two ways: either class imbalance correction was implemented using the SMOTE algorithm, or class imbalance correction was not implemented. A random search was conducted using the default parameters of the randomForest package (R statistical software version 3.5.2) for hyperparameter tuning: number of trees in the forest (=ntree): sequential number from 25 to 200 in increments of 25; the number of variables randomly sampled as candidates at each split (=mtry): sequential number from 1 to 5 in increments of 1; weights used only in sampling data to grow each tree: none; minimum size of terminal nodes (=nodesize): 1; the maximum number of terminal nodes trees in the forest can have (=maxnodes): trees are grown to the maximum possible; the importance of predictors not assessed.

### 2.6. External Validation of Constructed Radiomics Model

The constructed RF models were applied to the external test set. Three independent radiologists reviewed the external test set to compare the diagnostic performance of the RF model and the radiologists and evaluate the clinical usefulness. Two radiologists (S.-Y.L. and S.K., with 12 years and 1 year of experience in musculoskeletal radiology, referred to as R1 and R2, respectively) and a first-year resident in training (Y.-J.H, R3) reviewed CT images using two steps. In the first step, to compare the diagnostic performance between the RF model and the radiologists, they reviewed the CT images without assistance from the model. The images were reviewed three months after patient recruitment and VOI selection to avoid recall bias. All radiologists scored each lesion as MM or metastasis using a 5-point scale (0, definite MM; 1, probable MM; 2, equivocal; 3, probable osteolytic metastasis; 4, definite osteolytic metastasis). In the second step, to evaluate clinical utility, the radiologists reviewed CT images with the assistance of the RF model results. The RF model used in the second step was the only model without case imbalance correction. CT images were re-evaluated by the same radiologists two months after the first review to avoid recall bias. All radiologists were given information about the diagnostic performance of the RF model (sensitivity, specificity, accuracy, and AUC). The results of the RF model predicting the likelihood of MM for each case were provided as a probability score. All radiologists scored each lesion using the above-mentioned 5-point scale. They were blinded to clinical information in both steps. The overall workflow of the radiomics model development and validation is displayed in Figure 3.

### 2.7. Statistical Analysis

The metric for the RF model’s performance was the area under the curve (AUC) for the training and external test sets. Confidence intervals from 2000 bootstrapped samples were acquired for AUC. To calculate the sensitivity, specificity, and accuracy, a score of 0–2 points was assigned as MM, and a score of 3–4 points was assigned as metastasis. The AUC of the RF model was compared with the values of the three radiologists using DeLong’s test [23]. The sensitivity, specificity, and accuracy of the RF model, using a threshold calculated from Youden’s index [24], and the radiologists were compared using McNemar’s test. The AUC values of the first and second steps of the review were compared for each radiologist using DeLong’s test [23], while the sensitivity, specificity, and accuracy were compared using McNemar’s test.

Model development and statistical analysis, as well as feature reduction, were performed by an experienced statistician (J.L.) using commercially available software (MedCalc Statistical Software version 19.2.1, MedCalc Software Ltd., Ostend, Belgium; or R statistical software version 3.5.2).

## 3. Results

### 3.1. Patient Characteristics

A total of 175 patients (MM, n = 49) were included in the training set, while 50 patients (MM, n = 15) were included in the external test set. In total, there were 425 lesions (MM, n = 108) in the training set and 85 lesions (MM, n = 25) in the external test set. The most common primary malignancy of patients with osteolytic metastases was lung cancer, followed by breast cancer. Table 1 shows detailed patient demographics.

### 3.2. Radiomics Feature Selection and Model Development

A total of 1218 radiomics features were extracted, 380 of which were excluded because they were not reproducible; thus, 838 features were finally selected. The elastic net selected 71 radiomics features as input for RF model generation, with optimal parameters (λ = 0.01851 and α = 0.512). Table 2 shows the top 10 features according to importance. The AUC values (with a 95% confidence interval) of the radiomics model were 0.807 (0.759–0.854) without case imbalance correction and 0.821 (0.775–0.868) with case imbalance correction.

### 3.3. Diagnostic Performance of Radiomics Model on External Test Set

The AUC values of the radiomics models were 0.842 (0.752–0.932) and 0.762 (0.648–0.876) with and without data imbalance correction, respectively (Table 3). Conversely, using a cutoff of 0.24 for probability, the sensitivity, specificity, and accuracy values were 0.60 (0.39–0.79), 0.82 (0.75–0.94), and 0.75 (0.58–0.96), respectively. During the first step of the review, the sensitivity, specificity, accuracy, and AUC values of radiologists varied according to experience. The specificity of the radiomics model was significantly superior to that of the three radiologists (*p* ≤ 0.016). The sensitivity of the radiomics model was inferior to that of the three radiologists, although statistical significance was found for only one radiologist (*p* = 0.022). The AUC of the radiomics model was not significantly different from that of all radiologists (*p* ≥ 0.179) (Figure 4). During the second step of the review, the specificity was significantly increased for two radiologists (*p* = 0.004 for R2, *p* = 0.001 for R3), and the sensitivity was increased for all three radiologists but without statistical significance (*p* ≥ 0.500) compared to the first step. The AUC values of the second step were significantly superior to those of the first step for all three radiologists (*p* < 0.001) (Figure 5 and Table 4).

## 4. Discussion

Radiomics analysis was shown to have the potential to differentiate MM and osteolytic bone metastases on contrast-enhanced CT images. We found that the diagnostic performance of a radiomics-based machine-learning model was as good as that of an experienced radiologist. Furthermore, adding radiomics analysis increased the specificity for inexperienced radiologists. Finally, our study demonstrates that adding radiomics-based analysis increased the diagnostic performance of all three participating radiologists. This may be because metastases are more heterogeneous than MM on images. However, the radiologists’ accuracy in the current study was lower than that in the previous study (48.2–65.9% vs. 71.8–74.7%) for conventional CT readings [8]. The reason for the decreased accuracy in the current study is that we lacked an evaluation of intertumoral homogeneity, as only single lesions were assessed. 

Radiomics is a quantitative analysis technique for medical imaging that has been actively studied in recent years. Studies have been published on detecting and distinguishing bone lesions using CT-based radiomics analysis, and most of them are based on tumors [25,26,27]. A CT–radiomics model that can distinguish bone metastases from normal bone marrow was recently published [25]. They used automated lesion segmentation after manually locating lesions to collect sufficient data. On the contrary, we manually segmented lesions [26,27], which requires more time and effort but is more accurate. CT–radiomics models that differentiate malignant and benign bone tumors, atypical cartilaginous tumors, and chondrosarcomas were also developed recently [26,27]. This is the first study to investigate the feasibility of a CT–radiomics machine learning model for differentiating MM and metastases. A recent investigation of ^18^F-FDG PET/CT images reported two classification models using CT and PET features as inputs. Similar to our study, the AUC of a multivariate logistic regression model using radiomics features from CT images was comparable to that of human experts (0.897 vs. 0.840, *p* = 0.229). Conversely, the AUC values in our study were inferior to those in a study employing PET/CT, possibly because we only included cases of osteolytic metastasis. In general, MM has no osteoblast activity, so osteoblastic or bone metastasis with mixed density can be effectively distinguished from MM.

Interestingly, the radiomics model and the three radiologists showed conflicting sensitivity and specificity. Metastases can appear in various forms, and there are no specific imaging findings. Since MM is associated with specific imaging findings, an osteolytic lesion will be diagnosed as MM in the presence of these features. However, it is difficult to exclude MM in cases without these specific findings. This is where a radiomics model would be useful. With nonspecific osteolytic lesions, the finding of metastasis in the radiomics model would allow a confident diagnosis. Though it likely cannot replace biopsy, the presented radiomics model can help to differentiate MM and osteolytic bone metastases.

The training set included about twice as many metastatic lesions as MM. This data imbalance probably resulted in a model biased toward one class. Models in which the data imbalance was modified showed better diagnostic performance in both training and external validation sets than unmodified models. However, artificially balanced data may not represent the true population and could potentially eliminate useful information about the data itself. It could also result in overfitting of the model.

Our study has a few limitations. There may have been selection bias because the study data were collected retrospectively, and only the largest lesions were included for analysis. The evaluation of radiomics features was conducted only with a single software program. We only used an RF method without testing other classifiers. The data of this study partially overlap those of our previous study [8]. Data for this study were collected retrospectively from different CT scanners at different clinical institutions. CT parameters, including radiation dose and reconstruction kernel, may affect radiomic features [28]. However, the CT parameters in this study showed small differences between scanners because all images were collected with widely used protocols. In addition, resampling to isometric voxels corrected the variability related to section thickness [17].

The radiomics model showed equivalent diagnostic performance to that of radiologists. The model can improve the work of inexperienced radiologists in differentiating MM on CT scans in patients with osteolytic bone tumors. In conclusion, CT–radiomics analysis can enable the differentiation of MM and bone metastases.

## Figures and Tables

**Figure 1 diagnostics-13-00755-f001:**
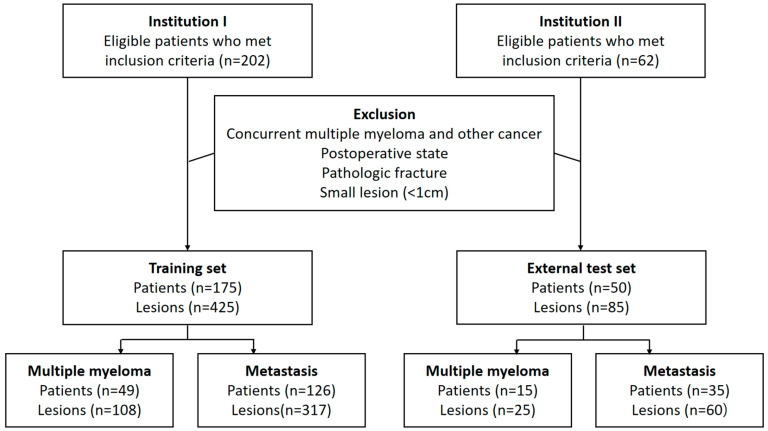
Flowchart showing inclusion and exclusion criteria of patients.

**Figure 2 diagnostics-13-00755-f002:**
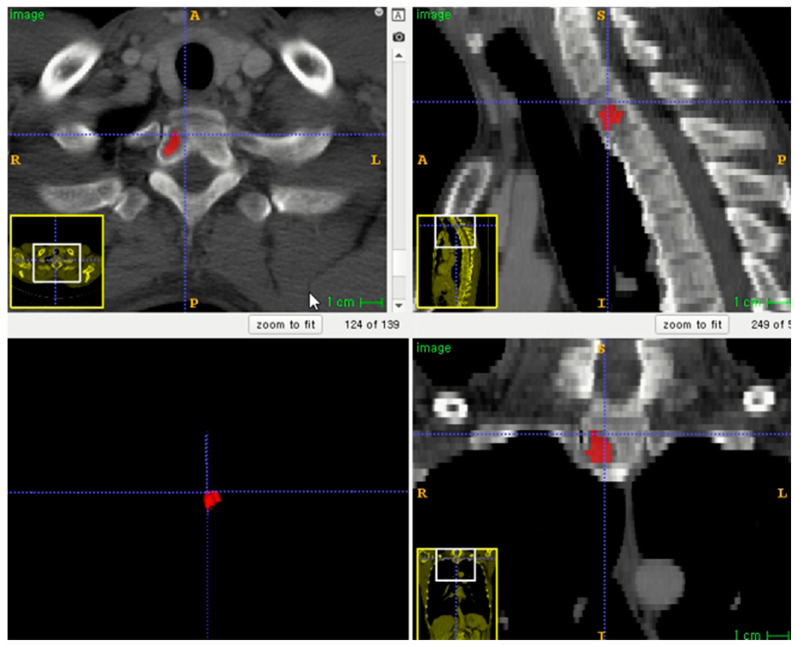
Example of lesion segmentation on axial skeleton.

**Figure 3 diagnostics-13-00755-f003:**
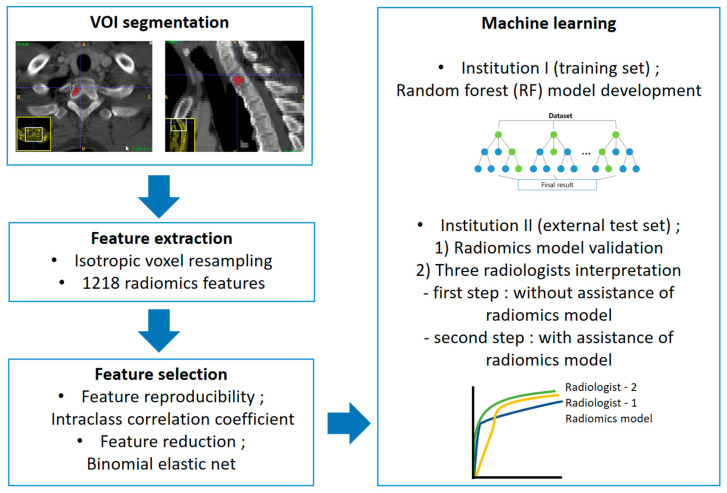
Radiomics pipeline of this study.

**Figure 4 diagnostics-13-00755-f004:**
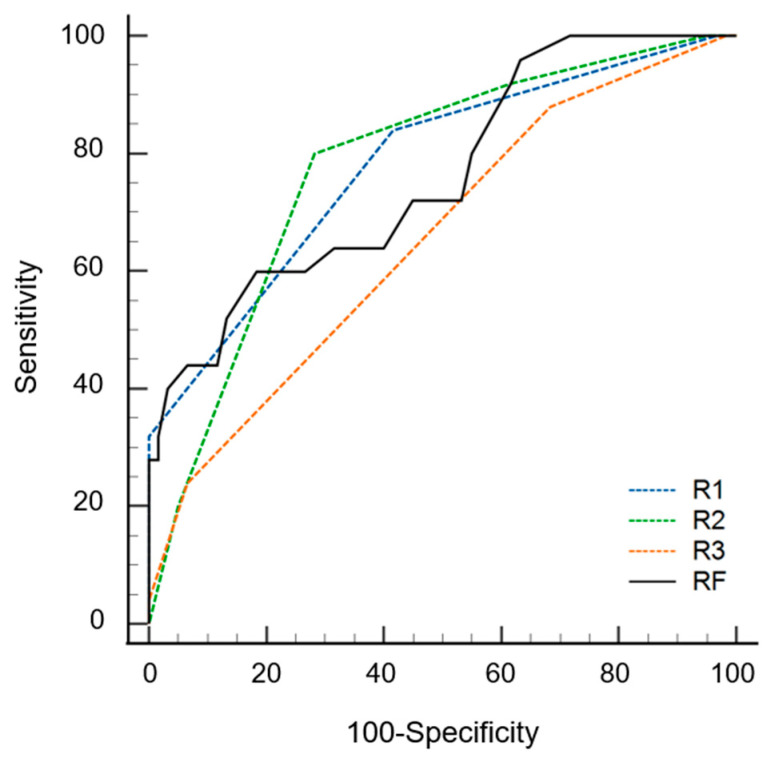
Receiver operating curves of radiomics model (RF) and three radiologists (R1, R2 and R3) for differentiating multiple myeloma from osteolytic metastasis.

**Figure 5 diagnostics-13-00755-f005:**
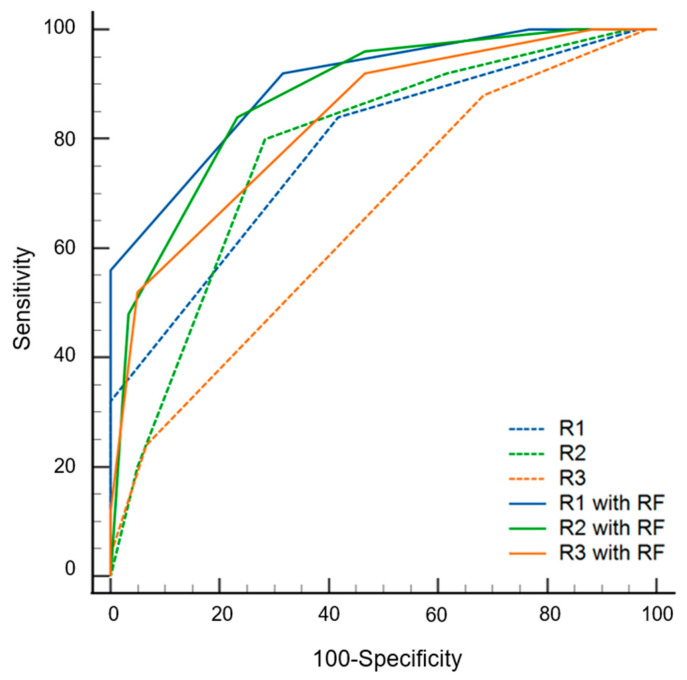
Diagnostic performance of three radiologists (R1, R2 and R3) with and without the assistance of radiomics model (RF) results.

**Table 1 diagnostics-13-00755-t001:** Patient characteristics.

	Training Set		External Test Set	
Characteristics	MM	Osteolytic Metastases	MM	Osteolytic Metastases
Number of patients	49	126	15	35
Number of lesions	108	317	25	60
Age (mean ± SD; years)	60.7 ± 12.4	62.8 ± 13.1	74.8 ± 13.8	64.3 ± 11.4
Sex (M:F)	26:23	82:44	8:7	21:14
Primary origin (numbers of patients)	n/a	Breast cancer (15) Cervical cancer (3) Cholangiocarcinoma (4) Colorectal cancer (5) Endometrial cancer (1) Gallbladder cancer (2) Hepatocellular carcinoma (17) Lung cancer (50) Melanoma (1) Neuroendocrine tumor (2) Ovarian cancer (1) Pancreatic cancer (1) Prostate cancer (3) Renal cell carcinoma (10) Stomach cancer (5) Thyroid cancer (3) Undifferentiated spindle cell sarcoma (1) Urothelial cell carcinoma (1)	n/a	Breast cancer (4) Colorectal cancer (1) Endometrial cancer (1) Gallbladder cancer (1) Hepatocellular carcinoma (2) Lung cancer (9)

MM, multiple myeloma.

**Table 2 diagnostics-13-00755-t002:** Top 10 radiomics features in terms of importance for differentiation between MM and osteolytic metastasis.

Radiomics Features	Importance
wavelet.LLL_gldm_DependenceNonUniformityNormalized	5.643140
wavelet.HLL_firstorder_Maximum	4.512872
wavelet.LLL_gldm_DependenceVariance	4.321726
wavelet.LHL_gldm_LargeDependenceEmphasis	3.974077
original_glcm_SumEntropy	3.792728
wavelet.LHL_glszm_SmallAreaEmphasis	3.693439
wavelet.LLL_firstorder_10Percentile	3.663920
original_gldm_DependenceVariance	3.499032
wavelet.LHH_firstorder_Kurtosis	3.353556
original_glcm_Imc2	3.305649

**Table 3 diagnostics-13-00755-t003:** Diagnostic performance of radiomics model.

Diagnostic Performance (AUC)	Radiomics Model without Class Imbalance Correction	Radiomics Model with Class Imbalance Correction
Training set	0.807 (0.759–0.854)	0.821 (0.775–0.868)
External test set	0.762 (0.648–0.876)	0.842 (0.752–0.932)

AUC, area under the receiver operating characteristic curve. Numbers within parentheses are 95% confidence interval ranges.

**Table 4 diagnostics-13-00755-t004:** Comparison of diagnostic performance of radiomics model and radiologists.

Diagnostic Performance	Sensitivity	Specificity	Accuracy	AUC
Radiomics model (A) ^†^	60.0% (15/25)	81.7% (49/60)	75.3% (64/85)	0.762 (0.648–0.876)
First review (B)				
R1	84.0% (21/25)	58.3% (35/60)	65.9% (56/85)	0.781 (0.681–0.881)
R2	92.0% (23/25)	38.3% (23/60)	54.1% (46/85)	0.778 (0.677–0.880)
R3	88.0% (22/25)	31.7% (19/60)	48.2% (41/85)	0.653 (0.545–0.762)
Second review (C)				
R1	92.0% (23/25)	68.3% (41/60)	75.3% (64/85)	0.900 (0.815–0.954)
R2	96.0% (24/25)	53.3% (32/60)	65.9% (56/85)	0.876 (0.786–0.937)
R3	92.0% (23/25)	53.3% (32/60)	64.7% (55/85)	0.833 (0.736–0.905)
Comparison of B and C (*p*-values)				
R1	0.500	0.109	0.039 *	<0.001 *
R2	1.000	0.004 *	0.002 *	<0.001 *
R3	1.000	0.001 *	0.001 *	<0.001 *

AUC, area under the receiver operating characteristic curve. Sensitivity, specificity, and accuracy were compared using McNemar’s test. AUCs were compared using DeLong’s test. † Threshold probability was 0.24 and calculated from Youden’s index. * *p* < 0.05.

## Data Availability

The datasets generated and/or analyzed during the current study are available from the corresponding author upon reasonable request.

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
