# Peer review of "Differentiating Multiple Myeloma and Osteolytic Bone Metastases on Contrast-Enhanced Computed Tomography Scans: The Feasibility of Radiomics Analysis"

_diagnostics, 2023, doi:10.3390/diagnostics13040755_

Round 1

Reviewer 1 Report

In this paper a radiomic analysis was proposed to differentiate multiple myeloma from osteolytic bone metastases using CT images. The performances of the radiomic model were compared to the performances of the medical experts, with and without radiomics assistance. The authors concluded that radiomics is feasible for the specific context and that it can improve human performances if adopted as radiologist’s assistant.

Although radiomics is currently adopted in several different contexts, it seems that its application to differentiate multiple myeloma and bone metastases was faced considering MRI only, and not CT; thus, the proposed application is innovative. I think that the most interesting part of this work is the evaluation of its usefulness when adopted as clinician’s assistant. In general the manuscript is well written and organized.

However, I believe that the authors should consider some methodological issues that could improve the methodological robustness of the work.

11.       The considered dataset is composed by images acquired by two different clinical institutes and three different scanners. The authors should assess if batch effects are present in the features distribution and consider to appropriately harmonize them (maybe using Combat strategy).

22.       Both datasets present high class imbalance, since the number of osteolytic metastases is more than twice larger than multiple myeloma lesions. I suggest to implement a class imbalance correction during the training procedure (e.g. using SMOTE algorithm or similar).

33.       For the same reason, it is not surprising that the choice of a probability threshold equal to 0.5 leads to very low sensitivity. In fact, the optimal threshold calculated by Youden’s index is lower than 0.5 and gives better results. I suggest reporting the optimal threshold only and deleting the default threshold of 0.5 from the results.

44.       The radiomic model and the three radiologists presented opposite performances in terms of sensitivity and specificity: the radiomic model favored the specificity (and thus had a lower number of osteolytic metastases misclassified as MM), whilst clinicians favored the sensitivity (with a lower number of MM misclassified as osteolytic metastases). I suggest the authors to discuss this point in the discussion, by also better enhancing which condition is preferable in the clinical practice.

Author Response

In this paper a radiomic analysis was proposed to differentiate multiple myeloma from osteolytic bone metastases using CT images. The performances of the radiomic model were compared to the performances of the medical experts, with and without radiomics assistance. The authors concluded that radiomics is feasible for the specific context and that it can improve human performances if adopted as radiologist’s assistant.

Although radiomics is currently adopted in several different contexts, it seems that its application to differentiate multiple myeloma and bone metastases was faced considering MRI only, and not CT; thus, the proposed application is innovative. I think that the most interesting part of this work is the evaluation of its usefulness when adopted as clinician’s assistant. In general the manuscript is well written and organized.

However, I believe that the authors should consider some methodological issues that could improve the methodological robustness of the work.

  1. The considered dataset is composed by images acquired by two different clinical institutes and three different scanners. The authors should assess if batch effects are present in the features distribution and consider to appropriately harmonize them (maybe using Combat strategy).

: We agree with Reviewer 1. The discussion on this was inserted as a limitation.

<Lines 305-310 (Discussion)>

Data of this study retrospectively were collected from different CT scanners of different clinical institutes. CT parameters including radiation dose and reconstruction kernel may affect radiomic features [28]. However, the CT parameters of this study showed small difference between CT scanners because all images collected with general widely used protocols. In addition, resampling to isometric voxels corrected the variability related to section thickness [17].

  1. Both datasets present high class imbalance, since the number of osteolytic metastases is more than twice larger than multiple myeloma lesions. I suggest to implement a class imbalance correction during the training procedure (e.g. using SMOTE algorithm or similar).

: Class imbalance was not corrected because it reflects the actual prevalence rate of multiple myeloma and metastases. However, the results with class imbalance correction can be useful information for readers as well. Then results and discussion of applying SMOTE algorithm were added to the manuscript.

<Lines 152-154 (Methods)>

The RF models were developed in two ways: class imbalance correction using SMOTE algorithm was implemented and class imbalance correction was not implemented.

<Lines 217-219, 224-225 (Results)>

… model development…

The AUC values (with 95% confidence intervals) of the radiomics model were 0.807 (0.759–0.854) without case imbalance correction, and 0.821 (0.775-0.868) for with case imbalance correction, respectively.

… radiomics model for the external test set…

The AUC values of the radiomics models were 0.842 (0.752-0.932) and 0.762 (0.648–0.876), with and without data imbalance correction (Table 3).

<Lines 295-300 (Discussion)>

The lesions included in the training set numbered about twice as many metastases as in MM. This data imbalance probably resulted in a model biased toward one class. Models involving modification of data imbalance showed better diagnostic performance in both training and external validation sets than unmodified models. However, artificially balancing the data may not represent the true population and have the potential to eliminate useful information about the data itself. Also, it can result in overfitting of the model.

  1. For the same reason, it is not surprising that the choice of a probability threshold equal to 0.5 leads to very low sensitivity. In fact, the optimal threshold calculated by Youden’s index is lower than 0.5 and gives better results. I suggest reporting the optimal threshold only and deleting the default threshold of 0.5 from the results.

: All results with the default threshold of 0.5 were removed

  1. The radiomic model and the three radiologists presented opposite performances in terms of sensitivity and specificity: the radiomic model favored the specificity (and thus had a lower number of osteolytic metastases misclassified as MM), whilst clinicians favored the sensitivity (with a lower number of MM misclassified as osteolytic metastases). I suggest the authors to discuss this point in the discussion, by also better enhancing which condition is preferable in the clinical practice.

: We added the issues in the discussion as follows:

<Lines 287-294 (Discussion)>

Interestingly, the radiomics model and three radiologists showed conflicting sensitivity and specificity. Metastasis can appear in various forms, and there are no specific imaging findings. Since MM is associated with specific imaging findings, an osteolytic lesion is diagnosed as MM in the presence of these features. However, it is difficult to exclude MM in cases without these specific findings. This is where a radiomics model would be useful. In non-specific osteolytic lesions, findings of metastasis in the radiomics model allow confident diagnosis. Though it likely cannot replace biopsy, the presented radiomics model helps differentiate MM and osteolytic bone metastases.

Reviewer 2 Report

The main novelty of the manuscript is the application of radiomics analysis to mutliple myeloma and osteolytic bone metastases in CT. The paper is well-written and offers the details about datasets and a radiomics analysis pipeline (however, it would be perfect if the data and/or code were published as well, for full reproducibility). I have a couple of comments but my main doubts are connected with the claim that the radiomics model improves the radiologists' performance. The details are given in the comments below. 

- I feel like some parts of the Discussion should be moved to Introduction, especially all the descriptions of related papers and their results, i.e.:

 - "Our previous research which attempted to differentiate MM and osteolytic metastases on contrast-enhanced CT images found that intratumoral and intertumoral homogeneity were key image findings for MM [21]"

 - a large part of the section between lines 271-282

 - Information about results for PET

Please rearrange that.

- I miss some more details about the Random Forest classifier used, i.e., the number of trees, max depth

- Major: I miss the description of how the assistance of radiomics model was implemented during evaluation. What was displayed to the radiologist and how was it presented? A binary prediction of the model? A probability score? Also, was the second step of the evaluation performed right after the first step? The increased performance of the radiologists might be connected not only with the support from classifier but also with the initial familiarization with all the tumors in the first step (suprisingly, the authors try to avoid a recall bias between images review and evaluation as mentioned in line 156, but they do not address a similar recall bias between first and second step of evaluation). Without proper disentanglement of these effects, the results are not reliable. If so, this must be clearly stated in the article.

- I see many inconsistencies in the number of digits and formatting of the results, i.e. in Table 2, in line 207, in Table 3. In Table 3, 0-1 range is used for some metrics and in Table 4 0-100% range is used. 

Author Response

The main novelty of the manuscript is the application of radiomics analysis to mutliple myeloma and osteolytic bone metastases in CT. The paper is well-written and offers the details about datasets and a radiomics analysis pipeline (however, it would be perfect if the data and/or code were published as well, for full reproducibility). I have a couple of comments but my main doubts are connected with the claim that the radiomics model improves the radiologists' performance. The details are given in the comments below.

- I feel like some parts of the Discussion should be moved to Introduction, especially all the descriptions of related papers and their results, i.e.:
- "Our previous research which attempted to differentiate MM and osteolytic metastases on contrast-enhanced CT images found that intratumoral and intertumoral homogeneity were key image findings for MM [21]"
 - a large part of the section between lines 271-282
 - Information about results for PET
Please rearrange that.

: All of them have been rearranged. All results from previous studies were moved to Introduction,

<Lines 63-78 (introduction)>

Our previous research which attempted to differentiate MM and osteolytic metastases on contrast-enhanced CT images found that intratumoral and intertumoral homogeneity were key image findings for MM [8]. Another study performing histogram analysis of diffusion-weighted MRI scans revealed that the apparent diffusion coefficient of MM was more narrowly distributed than that of metastases [9]. In addition, several studies using CT and MR images confirmed the difference in homogeneity between MM and bone metastases. Mutlu et al. revealed that osteolytic bone metastases showed more high-density areas and more significant heterogeneity than MM [10]. Radiomics analysis allows objective evaluation of tumor heterogeneity and may provide new information for imaging differentiation of tumors [11-13]. A recent MRI radiomics study of Xiong et al. revealed that a machine-learning model developed from radiomics features of T1- and T2-weighted images successfully differentiated MM and osteolytic bone metastases [14]. Another investigation of 18F-FDG PET/CT images found that the radiomics model successfully classified MM and bone metastases with significantly improved diagnostic performance compared to both the human experts and the conventional PET parameter SUVmax [15].

- I miss some more details about the Random Forest classifier used, i.e., the number of trees, max depth

: The hyperparameters were added.

<Lines 154-161 (Methods)>

Random search with the default parameters were used from “randomForest” package (R statistical software version 3.5.2) for hyperparameter tuning; Number of trees in the forest (=ntree): sequential number from 25 to 200 with increments of 25, Number of variables randomly sampled as candidates at each split (=mtry): sequential number from 1 to 5 with increments of 1, Weights used only in sampling data to grow each tree: None, Minimum size of terminal nodes (=nodesize): 1, Maximum number of terminal nodes trees in the forest can have (maxnodes): trees are grown to the maximum possible, Importance of predictors not assessed

- Major: I miss the description of how the assistance of radiomics model was implemented during evaluation. What was displayed to the radiologist and how was it presented? A binary prediction of the model? A probability score? Also, was the second step of the evaluation performed right after the first step? The increased performance of the radiologists might be connected not only with the support from classifier but also with the initial familiarization with all the tumors in the first step (suprisingly, the authors try to avoid a recall bias between images review and evaluation as mentioned in line 156, but they do not address a similar recall bias between first and second step of evaluation). Without proper disentanglement of these effects, the results are not reliable. If so, this must be clearly stated in the article.

: We have added missing details.

<Lines 173-181 (Method)>

In the second-step for the evaluation of clinical utility, radiologists reviewed CT images with assistance of the results from the RF model. CT images were re-evaluated by the same radiologists two months after the first review to avoid recall bias. All radiologists were given information about the diagnostic performance of the RF model (sensitivity, specificity, accuracy, and AUC). In addition, the results of the RF model predicting the likelihood of MM for each case were provided as probability score to radiologists. All radiologists scored each lesion using the above mentioned five-point scale. Clinical information was blinded to radiologists in both steps.

- I see many inconsistencies in the number of digits and formatting of the results, i.e. in Table 2, in line 207, in Table 3. In Table 3, 0-1 range is used for some metrics and in Table 4 0-100% range is used.

: They were all modified in a consistent manner.

Reviewer 3 Report

Manuscript is very good structured, materials and methods are well described and thus enable independent confirmation. Results are significant to clinical practice with important everyday application. Literature and discussion should be improved with recent literature describing software for detection of different bone lesions on CT.

Author Response

Manuscript is very good structured, materials and methods are well described and thus enable independent confirmation. Results are significant to clinical practice with important everyday application. Literature and discussion should be improved with recent literature describing software for detection of different bone lesions on CT.

: We have added discussion with recent literatures

<Lines 269- 277 (Discussion)>

Radiomics is a quantitative analysis technique for medical imaging that has been actively studied in recent years. Studies have been published to detect and distinguish bone lesions using CT-based radiomics analysis, and most of them are based on tumors [25-27]. A CT-radiomics model that can distinguish bone metastases from normal bone marrow recently has been published [25]. They used automated lesion segmentation after manually locating lesions to collect sufficient data. On the contrary, we manually segmented lesions [26, 27], which requires more time and effort but is more accurate. CT-radiomics models that differentiate malignant and benign bone tumors, atypical cartilaginous tumors, and chondrosarcomas also were developed recently [26, 27].

  1. Naseri, H.; Skamene, S.; Tolba, M.; Faye, M. D.; Ramia, P.; Khriguian, J.; Patrick, H.; Andrade Hernandez, A. X.; David, M.; Kildea, J. Radiomics-based machine learning models to distinguish between metastatic and healthy bone using lesion-center-based geometric regions of interest. Scientific reports2022, 12, 9866.
  2. Sun, W.; Liu, S.; Guo, J.; Liu, S.; Hao, D.; Hou, F.; Wang, H.; Xu, W. A CT-based radiomics nomogram for distinguishing between benign and malignant bone tumours. Cancer imaging 2021, 21, 20.
  3. Gitto, S.; Cuocolo, R.; Annovazzi, A.; Anelli, V.; Acquasanta, M.; Cincotta, A.; Albano, D.; Chianca, V.; Ferraresi, V.; Messina, C., et al. CT radiomics-based machine learning classification of atypical cartilaginous tumours and appendicular chondrosarcomas. EBioMedicine2021, 68, 103407.

Round 2

Reviewer 1 Report

I thank the authors for their replies to my comments and for having addressed my suggestions. I think that the manuscript is now improved, with more robust results and methods described in more details.

I have two minor comments:

·       It is not clear to me which model (with or without unbalance correction) has been adopted for the second evaluation by the readers. Please, add this information

·         I suggest the authors to deeply revise the text, especially the novel addictions, since I have found some typos or grammar errors.

Author Response

I thank the authors for their replies to my comments and for having addressed my suggestions. I think that the manuscript is now improved, with more robust results and methods described in more details.

I have two minor comments:

It is not clear to me which model (with or without unbalance correction) has been adopted for the second evaluation by the readers. Please, add this information

Author’s Reply:

We added the sentence “The RF model used in the second step was the only model without case imbalance correction.” in METHODS section (lines 175-176). 

I suggest the authors to deeply revise the text, especially the novel addictions, since I have found some typos or grammar errors.

Moderate English changes required

Author’s Reply:

After carefully checking the entire text, revising the sentence, and requesting the entire manuscript to MDPI English Editing company to correct the English language.

Reviewer 2 Report

Authors addressed all my comments

Author Response

Authors addressed all my comments

Author's reply:

We appreciated all of constructive questions and comments, as to improving our 
manuscript